# SReNet: Spectral Refined Network for Solving Operator Eigenvalue Problem

## Abstract

Solving operator eigenvalue problems helps analyze intrinsic data structures and relationships, yielding substantial influence on scientific research and engineering applications. Recently, novel approaches based on deep learning have been proposed to obtain eigenvalues and eigenfunctions from the given operator, which address the efficiency challenge arising from traditional numerical methods. However, when solving top-$L$ eigenvalues problems, these learning-based methods ignore the information that could be inherited from other known eigenvectors, thus resulting in a less-than-ideal performance. To address the challenge, we propose the **S**pectral **Re**fined **Net**work (**SReNet**). Our novel approach incorporates the power method to approximate the top-$L$ eigenvalues and their corresponding eigenfunctions. To effectively prevent convergence to previous eigenfunctions, we introduce the Deflation Projection that significantly improves the orthogonality of the computed eigenfunctions and enables more precise prediction of multiple eigenfunctions simultaneously. Furthermore, we develop the adaptive filtering method that dynamically leverages intermediate approximate eigenvalues to construct rational filters that filter out predicted eigenvalues, when predicting the successive eigenvalue of the given problem. During the iterative solving, the spectral transformation is performed based on the filter function, converting the original eigenvalue problem into an equivalent problem that is easier to converge. Extensive experiments demonstrate that our approach consistently outperforms existing learning-based methods, achieving state-of-the-art performance in accuracy.

## 1 Introduction

The operator eigenvalue problem is a prominent focus in many scientific fields (Elhareef & Wu, 2023; Buchan et al., 2013; Cuzzocrea et al., 2020; Pfau et al., 2023) and engineering applications (Diao et al., 2023; Chen & Chan, 2000), where eigenvalues are commonly used to analyze the fundamental geometric structures and relationships within data (Markovsky, 2012; Blum et al., 2020). Traditional methods that solve operator eigenvalue problems typically involve two steps. First, they apply numerical discretization methods to transform the operator into a matrix, such as Finite Element Methods (FEM) (LeVeque, 2002). Then numerical linear algebra techniques are employed to solve the eigenvalues and eigenvectors of the given matrix, utilizing methods like Krylov-Shur (Watkins, 2007; Liesen & Strakos, 2013) and the Locally Optimal Block Preconditioned Conjugate Gradient (LOBPCG) algorithm (Knyazev, 2001). These methods iteratively generate a subspace that approximates the invariant subspace of the matrix, allowing the original problem to be solved within this subspace. However, traditional numerical methods are constrained by the curse of dimensionality, as the computational complexity increases quadratically or even cubically with the size of the matrix (Watkins, 2007). Furthermore, storing the iterative subspaces incurs significant memory requirements when solving high-dimensional problems (Stewart, 2002).

A promising alternative is using neural networks to approximate eigenfunctions (Pfau et al., 2018). These approaches replace the matrix representations with parametric nonlinear representations through neural networks. By designing appropriate loss functions, it updates parameters to approximate the desired operator eigenfunctions. These methods only require sampling specific regions without designing discretization grids, significantly reducing the algorithm design cost and helping mitigate unwanted approximation errors (He et al., 2022). Moreover, the parametric representation of neural networks offers stronger expressive power than linear matrix representations, requiring

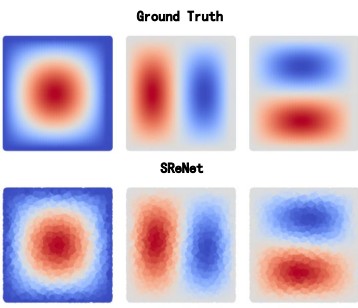 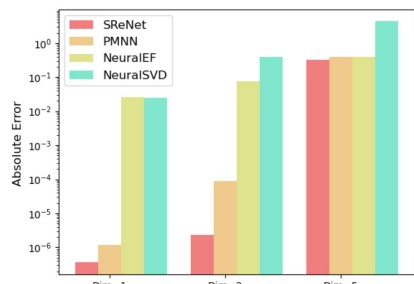

Figure 1: **Left**: Comparison of the eigenfunctions of the 2D Harmonic operator computed by SReNet and the Ground Truth. **Right**: Absolute error comparison of eigenvalues for the Schrödinger oscillator operator computed using various algorithms, the x-axis represents the operator dimension.

far fewer sampling points for the same problem compared to traditional methods (Nguyen et al., 2020). The memory overhead for these approaches depends only on the number of samples and the neural network parameters, eliminating the need for explicit matrix and iterative subspace storage, which significantly reduces memory costs in high-dimensional problems (Yang et al., 2023). Although these learning-based methods have been proposed to effectively solve operator eigenvalue problems, the numerical stability of these works is often influenced by the structure of the operator. The orthogonality of the predicted eigenfunctions and the spectral distribution of the operator directly determine the convergence rate of the iterations, thereby constraining the reduction of the loss function and ultimately affecting the accuracy of the solutions (Yang et al., 2023).

Inspired by the power methods (Golub & Van Loan, 2013), we propose a novel method, namely **S**pectral **Re**fined **Net**work (**SReNet**), that uses the neural network to predict the top-L eigenvalues and their eigenfunctions. SReNet takes the coordinates of sampling points as the input, while the outputs of SReNet are the eigenvalues and their eigenfunctions. SReNet employs the power method loss function as the guidance for iterative optimization. To prevent the predicted eigenfunctions from converging towards known invariant subspaces, we introduce the deflation projection into the loss function, which enhances the orthogonality and accuracy of the solved eigenfunctions. Furthermore, we develop adaptive filter techniques that utilize intermediate approximate eigenvalues to construct rational filters for transforming operator forms during the solution process, optimizing their spectral distribution (eigenvalue distribution) for higher solving efficiency. Extensive experiments demonstrate that SReNet significantly surpasses existing methods based on deep learning, achieving state-of-the-art performance.

In summary, our contributions are as follows:

- We introduce a novel learning-based method based on power method concepts for solving the top-L eigenvalues and their eigenfunctions of the differential operator.

- We employ the deflation projection that prevents convergence towards known invariant subspaces, enhancing the multi-eigenvalue problem-solving. We also develop adaptive rational filter techniques that utilize intermediate eigenvalues to accelerate the solution process and achieve higher solving efficiency.

- We conduct extensive experiments to evaluate the effectiveness of our proposed methods, achieving state-of-the-art precision across different operator eigenvalue problems.

## 2 PRELIMINARIES

### 2.1 OPERATOR EIGENVALUE PROBLEM

This paper primarily focuses on the eigenvalue problems of differential operators. Mathematically, an operator $\mathcal{L} : \mathcal{H}_1 \rightarrow \mathcal{H}_2$ is a mapping between two Hilbert spaces, $\mathcal{H}_1$ and $\mathcal{H}_2$ (Kantorovich & Akilov, 2014). If $\mathcal{L}$ is an operator acting on a Hilbert space $\mathcal{H}$ and $v(x) \in \mathcal{H}$, the eigenvalue

problem can be formulated as:

$$\mathcal{L}v = \lambda v, \tag{1}$$

where $v(x)$ is the eigenfunction associated with the eigenvalue $\lambda$ (Evans, 2022). The primary interest lies in determining the spectrum of operators, i.e., eigenvalues $\lambda$, and their corresponding eigenfunctions $v$ for $\mathcal{L}$.

Partial differential operators are common in operator eigenvalue problems, frequently encountered across scientific computing and engineering applications. Considering a self-adjoint operator $\mathcal{L}$ defined on a domain $\Omega \subset \mathbb{R}^D$, the operator eigenvalue problem can be expressed in the following form (Davies, 2007):

$$\begin{cases} \mathcal{L}v = \lambda v & \text{in } \Omega, \\ \mathcal{B}v = g & \text{on } \partial\Omega, \end{cases} \tag{2}$$

where $\Omega \subseteq \mathbb{R}^D$ serves as the domain, and $\mathcal{L}$ and $\mathcal{B}$ are differential operators acting within the interior and on the boundary of $\Omega$, respectively. The eigenpair $(v, \lambda)$ consists of $v(x)$, the eigenfunction associated with the operator $\mathcal{L}$, and $\lambda$, the corresponding eigenvalue. Typically, it is often necessary to solve for multiple eigenpairs, $(v_i, \lambda_i), i = 1, \ldots, L$ in applications such as obtaining energy basis functions from the Hamiltonian operator in quantum chemistry (Kittel & McEuen, 2018; Grosso & Parravicini, 2013; Hook & Hall, 2013) or modeling multiple acoustic modes (Shang, 1989; Mason, 2013; Thompson et al., 1991).

## 2.2 POWER METHOD

The power method is commonly used as the iterative algorithm for computing the largest eigenvalue and its corresponding eigenvector of the given matrix, particularly well-suited for large sparse matrices. The power method employs successive matrix-vector multiplications to progressively converge towards the largest eigenvector (Watkins, 2007). At the beginning, power method starts from a random and non-zero initial vector $\boldsymbol{x}^{(0)}$, and update the vector through the following iterative steps:

1. Iterative Update: Compute $\boldsymbol{y}^{(k)} = \boldsymbol{A}\boldsymbol{x}^{(k-1)}$.

2. Normalization: Normalize by setting $\boldsymbol{x}^{(k)} = \frac{\boldsymbol{y}^{(k)}}{\|\boldsymbol{y}^{(k)}\|}$.

3. Convergence Check: Repeat the process until $\boldsymbol{x}^{(k)}$ and $\boldsymbol{x}^{(k-1)}$ are close enough, or a predefined number of iterations is reached.

The matrix eigenvalue problem can be considered as the operator eigenvalue problem with a finite-dimensional linear operator. Assume $|\lambda_1| > |\lambda_2| \geq \ldots \geq |\lambda_n|$, where $\boldsymbol{v}_i$ is the eigenvector corresponding to $\lambda_i$. The power method constructs the following sequence:

$$\{\boldsymbol{x}^{(0)}, \boldsymbol{A}\boldsymbol{x}^{(0)}, \boldsymbol{A}^2\boldsymbol{x}^{(0)}, \ldots, \boldsymbol{A}^k\boldsymbol{x}^{(0)}, \ldots\}. \tag{3}$$

If

$$\boldsymbol{x}^{(0)} = a_1\boldsymbol{v}_1 + a_2\boldsymbol{v}_2 + \cdots + a_n\boldsymbol{v}_n \tag{4}$$

and $\boldsymbol{v}_1 \neq 0$, then

$$\boldsymbol{A}^k\boldsymbol{x}^{(0)} = a_1\lambda_1^k \left( \boldsymbol{v}_1 + \sum_{j=2}^{n} \frac{a_j}{a_1} \left( \frac{\lambda_j}{\lambda_1} \right)^k \boldsymbol{v}_j \right). \tag{5}$$

This formula indicates several factors that affect the convergence speed of the power method. First, the eigenvalue $\lambda_1$ in $\boldsymbol{A}^k\boldsymbol{x}^{(0)}$ is crucial. As iterations progress, the vector $\boldsymbol{x}^{(k)}$ increasingly aligns with $\boldsymbol{v}_1$. Secondly, the convergence rate is influenced by the ratio $|\lambda_1|/|\lambda_2|$; the larger the ratio, the faster the convergence. Additionally, the idea of the power method is to iteratively calculate $\boldsymbol{x}^{(k)}$ to $\boldsymbol{A}\boldsymbol{x}^{(k)}$, amplifying the impact of the largest eigenvalue $\lambda_1$ through actions of matrix $\boldsymbol{A}$, ultimately solving for $(\boldsymbol{v}_1, \lambda_1)$. Our algorithm is derived from this idea. For more details of the power method, we refer to Appendix B.1.

## 2.3 Deflation Projection

The deflation technique plays a critical role in solving eigenvalue problems, particularly when dealing with large-scale matrices or when multiple distinct eigenvalues need to be computed. Deflation projection is an effective deflation strategy that utilizes known eigenvalues and corresponding eigenfunctions to modify the structure of the matrix, thereby simplifying the computation of remaining eigenvalues (Saad, 2011; Kressner, 2005; Watkins, 2007; Arbenz et al., 2012).

The essence of deflation projection lies in constructing a projection matrix $\boldsymbol{P}$, typically $\boldsymbol{v}_1\boldsymbol{v}_1^\top$, where $\boldsymbol{v}_1$ is a known eigenvector. This projection matrix allows to modify the original matrix $\boldsymbol{A}$ to a new matrix $\boldsymbol{B} = \boldsymbol{A} - \lambda_1\boldsymbol{P}$. In matrix $\boldsymbol{B}$, the eigenvalue $\lambda_1$ corresponding to eigenvector $\boldsymbol{v}_1$ is effectively removed (or set to zero) from $\boldsymbol{A}$. We provide more details about deflation projection in the Appendix B.2.

## 2.4 Filtering Technique

The filtering technique is employed in numerical linear algebra to accelerate the solution of eigenvalue problems for large matrices (Saad, 2011). The core of this technique involves constructing suitable filter functions $F(\boldsymbol{A})$ to achieve a spectral transformation of the matrix $\boldsymbol{A}$. This spectral transformation helps optimize the spectral distribution of the matrix without altering the eigenvectors, thereby making the target eigenvalues more prominent in the transformed spectrum and easier to obtain (Watkins, 2007; Li et al., 2019).

The spectral transformation essentially applies a function transformation to the matrix. These functions, such as polynomials or rational functions, are designed to amplify the important part of the matrix (the eigenvalues we care about) and suppress the unnecessary ones (Fang & Saad, 2012; Winkelmann et al., 2019), thus we call it "filtering". The filter reduces the influence of irrelevant eigenvalues, making it easier to converge on the target eigenvalues (Miao, 2019; Miao & Wu, 2021). We provide more details about filtering technique in the Appendix B.3.

## 3 Method

### 3.1 Problem Formulation

We consider the operator eigenvalue problem for a differential operator $\mathcal{L}$ defined on a domain $\Omega \subset \mathbb{R}^D$. Our goal is to compute the top-$L$ eigenfunctions $v_i$ of $\mathcal{L}$, along with their corresponding eigenvalues $\lambda_i$, satisfying $\mathcal{L}v_i = \lambda_i v_i, i = 1, 2, \ldots, L$. We employ neural networks $NN_{\mathcal{L}}(\theta_i)$ parameterized by $\theta_i$. Each neural network maps the domain $\Omega$ into the real space $\mathbb{R}$, approximating the eigenfunctions $v_i$:

$$NN_{\mathcal{L}}(\cdot; \theta_i) : \Omega \to \mathbb{R}, \quad i = 1, 2, \ldots, L. \tag{6}$$

We discretize the domain by uniform random sampling $N$ point set

$$S \equiv \{\boldsymbol{x}_j = (x_j^1, \ldots, x_j^D) \mid \boldsymbol{x}_j \in \Omega, \, j = 1, 2, \ldots, N\}, \tag{7}$$

which makes up an $N \times D$ matrix $\boldsymbol{X}_{input}$. This matrix serves as the input to the neural networks $NN_{\mathcal{L}}(\theta_i)$. The networks output $L$ vectors $\boldsymbol{Y}_i \in \mathbb{R}^N$, representing the approximate values of the eigenfunctions $\tilde{v}_i(\cdot) = NN_{\mathcal{L}}(\cdot; \theta_i)$ at these sampled points:

$$\tilde{v}_i(\boldsymbol{x}_j) \equiv \boldsymbol{Y}_i(j), \quad i = 1, 2, \ldots, L, \quad j = 1, 2, \ldots, N. \tag{8}$$

The eigenvalues $\tilde{\lambda}_i$ are obtained by applying the operator $\mathcal{L}$ to the computed eigenfunctions $\tilde{v}_i$:

$$\tilde{\lambda}_i \equiv \frac{\tilde{v}_i^\top \mathcal{L} \tilde{v}_i}{\tilde{v}_i^\top \tilde{v}_i}, i = 1, 2, \ldots, L. \tag{9}$$

We iteratively update the neural network parameters $\theta_i$ using gradient descent, aiming to minimize the overall residual. The optimization problem is formulated as:

$$\min_{\theta_i \in \Theta} \frac{1}{N} \sum_{i=1}^{L} \sum_{j=1}^{N} [\tilde{v}_i(\boldsymbol{x}_j) - v_i(\boldsymbol{x}_j)]^2, \tag{10}$$

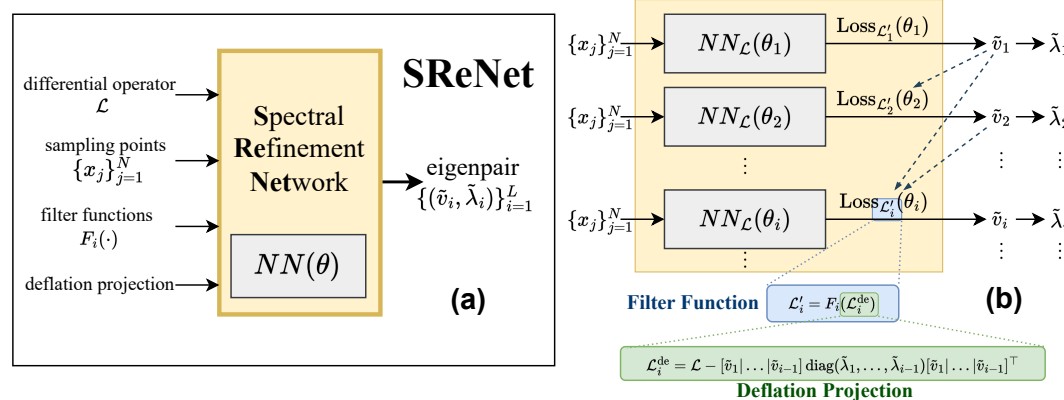

Figure 2: Overview of the **SReNet**. **(a)** Introduction to the inputs and outputs. **(b)** The SReNet comprises multiple neural networks, each tasked with predicting distinct eigenfunctions and eigenvalues. By employing **filter functions** and **deflation projection**, the algorithm integrates previously computed eigenfunctions and eigenvalues into the loss function.

where $\Theta$ denotes the parameter space of the neural networks. This approach does not require any training data, as it relies solely on satisfying the differential operator eigenvalue equations over the domain $\Omega$. However, the $v_i(\boldsymbol{x_j})$ is unknown, so we need to generate a $\tilde{u}_i^{k+1}(\boldsymbol{x_j})$ as the fitting of $v_i(\boldsymbol{x_j})$ by the output of the model and corresponding operator.

## 3.2 SPECTRAL REFINED NETWORK

SReNet learns to solve the operator eigenvalue problem by iterative optimization, which takes sampled points as the inputs, and outputs the eigenfunction. And the neural network in SReNet parameterized by parameters $\theta_i$ predicts the $i$-th eigenfunction $v_i$. To enhance the convergence speed and prediction accuracy, SReNet employs **Deflation Projection** and **Filter Function** for training, which are common spectral refinement techniques. Figure 2 shows the overview of SReNet.

Supposing that at the $k$-th iteration, the output of SReNet is $\boldsymbol{Y}_i^k(j)$ as the $i$-th eigenfunction, where $\theta_i^k$ represents the parameters of the neural network at $k$-th iteration,

$$\tilde{v}_i^k(x) \equiv NN_{\mathcal{L}}(x; \theta_i^k), x \in \Omega, \quad \tilde{v}_i^k(\boldsymbol{x_j}) = NN_{\mathcal{L}}(\boldsymbol{x_j}; \theta_i^k) = \boldsymbol{Y}_i^k(j), x_j \in S, j = 1, \ldots, N. \quad (11)$$

In particular, SReNet employs the Multi Layer Perceptron (MLP) as $NN_{\mathcal{L}}(\cdot; \theta_i)$ for eigenfunction prediction. Learning the $i$-th eigenfunction of a given operator requires updating the neural network to minimize the objection function in (10). Previous works, like (Yang et al., 2023), suggest that optimizing the loss function derived from the power method helps the eigenfunction prediction. The derivative loss function is defined as follows:

$$\text{Loss}_{\mathcal{L}}^{PM}(\boldsymbol{x_j}, \theta_i^k) = \frac{1}{N} \sum_{j=1}^{N} \left[ \tilde{v}_i^k(\boldsymbol{x_j}) - \tilde{u}_i^{k+1}(\boldsymbol{x_j}) \right]^2, \quad \tilde{u}_i^{k+1}(\boldsymbol{x_j}) = \frac{\mathcal{L}\tilde{v}_i^k(\boldsymbol{x_j})}{\|\mathcal{L}\tilde{v}_i^k(\boldsymbol{x_j})\|}. \quad (12)$$

For ease of reference, we use **PM Loss** to indicate the loss function in Eq (12). As suggested in Eq (12), we update the parameters of SReNet by iterative optimization. This is inspired by the power method that takes iterative steps to solve the eigenvalue problem. But we uses Automatic Differentiation (AD) to compute $\mathcal{L}\tilde{v}_i^k(\boldsymbol{x_j})$, which is the action of the operator $\mathcal{L}$ on $\tilde{v}_i^k$. For example, consider the operator $\mathcal{L}u = \alpha\Delta u + \beta \cdot \nabla u$, where $\alpha$ is a constant and $\beta$ is a constant vector. The operator $\mathcal{L}$ acting on the neural network output $\tilde{v}_i^k(x)$ can be expressed as:

$$\mathcal{L}\tilde{v}_i^k(x) = \alpha\Delta NN_{\mathcal{L}}(x; \theta_i^k) + \beta \cdot \nabla NN_{\mathcal{L}}(x; \theta_i^k), \quad (13)$$

where

$$\nabla NN_{\mathcal{L}}(x; \theta_i^k) = \left[ \frac{\partial NN_{\mathcal{L}}(x; \theta_i^k)}{\partial x^1}, \ldots, \frac{\partial NN_{\mathcal{L}}(x; \theta_i^k)}{\partial x^D} \right], \quad (14)$$

$$\Delta NN_{\mathcal{L}}(x; \theta_i^k) = \sum_{d=1}^{D} \frac{\partial^2 NN_{\mathcal{L}}(x; \theta_i^k)}{\partial x^{d^2}}. \tag{15}$$

Both the gradient $\nabla$ and the Laplacian $\Delta$ are computed using AD rather than numerical differentiation. This approach allows SReNet to effectively compute the $\mathcal{L}\tilde{v}_i^k(\boldsymbol{x_j})$.

Furthermore, as illustrated in Figure 2, we introduce **Deflation Projection** and **Filter Function** $F_i(\cdot)$ into the PM Loss to handle multiple eigenvalues problem and improve convergence speed. It is achieved by replacing the operator $\mathcal{L}$ used for calculating $\mathcal{L}\tilde{v}_i^k(\boldsymbol{x_j})$ in PM Loss. Specifically, the PM Loss for the $i$-th eigenfunction is $\text{Loss}_{\mathcal{L}_i'}^{PM}$, where

$$\mathcal{L}_i' = F_i(\mathcal{L}_i^{\text{de}}), \tag{16}$$

and

$$F_i(\mathcal{L}) \equiv (\mathcal{L} - \tilde{\lambda}_i I)^{-1}, \quad \mathcal{L}_i^{\text{de}} \equiv \mathcal{L} - Q_{i-1}\Sigma_{i-1}Q_{i-1}^{\top}. \tag{17}$$

Here, $Q_i$ and $\Sigma_i$ are operators representing the previously computed eigenfunctions and eigenvalues, $I$ is the identity operator and $\tilde{\lambda}_i$ is the current approximation of the $i$-th eigenvalue. In the following sections, we will provide a detailed explanation of the implementation of these two components. For details of SReNet, please see the pseudocode in Appendix C.

### 3.3 DEFLATION PROJECTION

PM Loss helps the neural networks converge to the eigenfunction associated with the largest eigenvalue of the given operator. However, it does not enforce orthogonality between different eigenfunctions, making it difficult to accurately compute multiple eigenfunctions.

Suppose we have already predicted $i-1$ eigenvalues $\tilde{\lambda}_1, \tilde{\lambda}_2, \ldots, \tilde{\lambda}_{i-1}$ and their corresponding eigenfunctions $\tilde{v}_1, \tilde{v}_2, \ldots, \tilde{v}_{i-1}$. To find the $i$-th eigenfunction, we aim to search within the residual subspace orthogonal to the previously computed eigenfunctions. To achieve this, we apply deflation projection to the operator $\mathcal{L}$:

$$\mathcal{L}_i^{\text{de}} \equiv \mathcal{L} - Q_{i-1}\Sigma_{i-1}Q_{i-1}^{\top}, \tag{18}$$

where

$$Q_{i-1} = [\tilde{v}_1 | \tilde{v}_2 | \ldots | \tilde{v}_{i-1}], \quad \Sigma_{i-1} = \text{diag}(\tilde{\lambda}_1, \tilde{\lambda}_2, \ldots, \tilde{\lambda}_{i-1}). \tag{19}$$

Here, the operator $Q_{i-1}$ includes previously computed eigenfunctions as its columns and can be considered a matrix in $\mathbb{R}^{N \times (i-1)}$. The operator $\Sigma_{i-1}$ represents a diagonal matrix of corresponding eigenvalues in $\mathbb{R}^{(i-1) \times (i-1)}$. This yields the corresponding loss function:

$$\text{Loss}_{\mathcal{L}_i^{\text{de}}}^{PM}(\boldsymbol{x_j}, \theta_i^k) = \frac{1}{N} \sum_{j=1}^{N} \left[ \tilde{v}_i^k(\boldsymbol{x_j}) - \tilde{u}_i^{k+1}(\boldsymbol{x_j}) \right]^2, \tilde{u}_i^{k+1}(\boldsymbol{x_j}) = \frac{\mathcal{L}_i^{\text{de}} \, \tilde{v}_i^k(\boldsymbol{x_j})}{\|\mathcal{L}_i^{\text{de}} \, \tilde{v}_i^k(\boldsymbol{x_j})\|}, \tag{20}$$

By employing the deflation projection, the gradient descent search space of the neural network is constrained to be orthogonal to the subspace spanned by $\{\tilde{v}_1, \tilde{v}_2, \ldots, \tilde{v}_{i-1}\}$. This approach prevents the neural network output $NN_{\mathcal{L}}(\theta_i)$ from converging to the invariant subspace formed by known eigenfunctions, thereby enhancing the orthogonality among the outputs of different neural networks $NN_{\mathcal{L}}(\theta_1), \ldots, NN_{\mathcal{L}}(\theta_{i-1})$. On one hand, this reduction in the search space accelerates the convergence toward the eigenfunctions $v_i$; On the other hand, it improves the orthogonality among the neural network outputs, which reduces the error in predicting the eigenfunction $\tilde{v}_i$.

In practice, we use the previously computed eigenvalues and eigenfunctions with the lowest approximation errors to construct the loss with deflation projection. This allows us to adaptively update the deflation operator, ensuring the method remains effective as more eigenfunctions are computed.

### 3.4 FILTER FUNCTION

In practice, we are typically interested in eigenvalues within a specific interval rather than solely the largest eigenvalue. Furthermore, the presence of larger eigenvalues can affect the convergence speed of the current eigenvalue calculations. These problems encourage us to employ another effective method, which enables SReNet to predict the eigenvalues we are interested at while suppressing the effects from large eigenvalues.

The filtering function is one of the choices, as it filters out the eigenvalues we don't need while amplifying the eigenvalues we want. In the iterative solving process of SReNet, we can obtain approximate eigenvalues $\tilde{\lambda}_i, i = 1, \ldots, L$. Using these approximate eigenvalues $\tilde{\lambda}_i$, we adaptively design filter functions $F_i(\cdot)$ to focus the operator's spectrum around the desired eigenvalue $\tilde{\lambda}_i$. By amplifying the magnitude of eigenvalue $\tilde{\lambda}_i$, the filter function $F_i(\cdot)$ accelerates the convergence speed of the neural network toward $\tilde{\lambda}_i$. The filter function $F_i(\cdot)$ can take various forms. In SReNet, to compute the $i$-th eigenpair $(v_i, \lambda_i)$, we adopt the following rational function form:

$$F_i(\mathcal{L}) \equiv (\mathcal{L} - \tilde{\lambda}_i I)^{-1}, \tag{21}$$

where $I$ is the identity operator. If we simultaneously apply deflation projection, the transformed operator $\mathcal{L}'_i$ can be expressed as:

$$\begin{aligned}
\mathcal{L}'_i = F_i(\mathcal{L}_i^{\text{de}}) &= F_i \left( \mathcal{L} - Q_{i-1} \Sigma_{i-1} Q_{i-1}^\top \right) \\
&= \left( \mathcal{L} - Q_{i-1} \Sigma_{i-1} Q_{i-1}^\top - \tilde{\lambda}_i I \right)^{-1}.
\end{aligned} \tag{22}$$

However, due to the presence of the inverse of the operator in the filter function $F_i(\mathcal{L})$, the transformed operator $\mathcal{L}'_i$ involves inverse computations, making it impossible to directly apply the previous power method loss template for forward iterations. To address this issue, we design the loss in the following inverse iteration form:

$$\text{Loss}_{\mathcal{L}'_i}^{PM}(\theta_i^k) = \frac{1}{N} \sum_{j=1}^{N} \left[ \tilde{v}_i^{k-1}(\boldsymbol{x_j}) - \tilde{u}_i^k(\boldsymbol{x_j}) \right]^2, \tag{23}$$

$$\tilde{u}_i^k(\boldsymbol{x_j}) = \frac{\mathcal{L}'_i \tilde{v}_i^k(\boldsymbol{x_j})}{\left\| \mathcal{L}'_i \tilde{v}_i^k(\boldsymbol{x_j}) \right\|}, \quad \mathcal{L}'_i = \mathcal{L} - Q_{i-1} \Sigma_{i-1} Q_{i-1}^\top - \tilde{\lambda}_i I. \tag{24}$$

By adaptively adjusting the parameters of the filter function based on known approximate eigenvalues, we can improve computational efficiency. This loss function effectively amplifies the eigenvalues near $\tilde{\lambda}_i$, which enhances the process of solving for the $i$-th eigenpair $(v_i, \lambda_i)$ .

When we do not use deflation projection to optimize the loss, we can also adopt the following form of the filter function:

$$F_i(\mathcal{L}) = \prod_{i_0=0}^{i-1} \left( \mathcal{L} - \tilde{\lambda}_{i_0} I \right) \cdot \left( \mathcal{L} - \tilde{\lambda}_i I \right)^{-1}. \tag{25}$$

The filter function can be modified according to specific requirements, such as using Chebyshev polynomial filters. In this paper, we primarily adopt the filter function in Eq (21) for our experiments.

## 4 EXPERIMENTS

We conducted comprehensive experiments to evaluate SReNet, focusing on:

- Solving top-$L$ operator eigenvalues in the Harmonic eigenvalue problem.
- Solving the principal eigenvalue in the Schrödinger oscillator equation.
- Solving zero eigenvalue in the Fokker-Planck equation.
- The ablation study.

**Baselines**: For this experiment, we selected three learning-based methods for computing operator eigenvalues as our baselines: 1. PMNN (Yang et al., 2023); 2. NeuralEF (Deng et al., 2022); 3. NeuralSVD (Ryu et al., 2024). For introductions to related work, see Appendix A.

**Experiment Settings**: To ensure consistency, all experiments were conducted under the same computational conditions. For further details on the experimental environment and parameters, please refer to Appendix D.

## 4.1 HARMONIC EIGENVALUE PROBLEM

Harmonic eigenvalue problems are common in fields such as structural dynamics and acoustics, and can be mathematically expressed as follows (Yang et al., 2023; Morgan & Zeng, 1998):

$$\begin{cases} -\Delta v = \lambda v, & \text{in } \Omega, \\ v = 0, & \text{on } \partial\Omega. \end{cases} \tag{26}$$

Here $\Delta$ denotes the Laplacian operator. We consider the domain $\Omega = [0,1]^D$ where $D$ represents the dimension of the operator, and the boundary conditions are Dirichlet. In this setting, the eigenvalue problem has analytical solutions, with eigenvalues and corresponding eigenfunctions given by:

$$\lambda_{n_1,\dots,n_D} = \pi^2 \sum_{k=1}^{D} n_k^2, \quad u_{n_1,\dots,n_D}(x_1,\dots,x_k) = \prod_{k=1}^{D} \sin(n_k \pi x_k), \quad n_k \geq 1. \tag{27}$$

To validate our algorithm's capability to compute top-$L$ eigenvalues in both low and high-dimensional settings, our experiments aim to calculate the first four eigenvalues of the Harmonic operator in $1, 2$ and $5$ dimensions. Since the PMNN model only computes the principal eigenvalue and cannot compute multiple eigenvalues simultaneously, it is not considered for comparison. NeuralEF, due to cumulative errors in its iterative orthogonalization process, experiences numerical instability in 2 and 5 dimensions, thus no data is available for these dimensions.

Table 1: Absolute error comparison for eigenvalues of Harmonic operators across algorithms. The first row lists the algorithms, the second row lists eigenvalue indexs and the first column lists the operator dimensions. The most accurate method is in bold.

| Method | NeuralEF | | | | NeuralSVD | | | | SReNet | | | |
|---|---|---|---|---|---|---|---|---|---|---|---|---|
| | $\lambda_1$ | $\lambda_2$ | $\lambda_3$ | $\lambda_4$ | $\lambda_1$ | $\lambda_2$ | $\lambda_3$ | $\lambda_4$ | $\lambda_1$ | $\lambda_2$ | $\lambda_3$ | $\lambda_4$ |
| Dim = 1 | 1.4e-1 | 2.9e+1 | 7.9e+1 | 1.4e+2 | 1.0e-1 | 4.1e+1 | 1.0e+0 | 1.4e+2 | **6.3e-10** | **1.7e+0** | **6.3e-1** | **1.6e+1** |
| Dim = 2 | - | - | - | - | 5.5e-2 | 2.1e-1 | 1.5e-1 | 2.6e+1 | **1.0e-5** | **3.0e-2** | **6.8e-2** | **1.0e-1** |
| Dim = 5 | - | - | - | - | 2.5e-1 | 2.9e+1 | 2.9e+1 | 2.9e+1 | **2.3e-4** | **9.5e-5** | **6.2e-5** | **1.3e-3** |

Table 2: Residual comparison for eigenpairs of SReNet and NeuralSVD for solving 5-dimensional Harmonic operator eigenvalue problems. The first row indicates the eigenpair index.

| Index | $(v_1, \lambda_1)$ | $(v_2, \lambda_2)$ | $(v_3, \lambda_3)$ | $(v_4, \lambda_4)$ |
|---|---|---|---|---|
| NeuralSVD | 5.924e+0 | 5.920e+0 | 5.921e+0 | 5.920e+0 |
| SReNet | 4.864e-4 | 3.060e-3 | 5.980e-3 | 4.447e-3 |

Firstly, as demonstrated in Table 1, SReNet significantly outperforms existing methods across all tasks, with precision improvements reaching up to nine orders of magnitude. This enhancement primarily stems from the deflation projection. It effectively excludes solved invariant subspaces during the multi-eigenvalue solution process, thereby preserving the accuracy of multiple eigenvalues. This strongly validates the efficacy of our algorithm.

Secondly, in 5 dimension, SReNet consistently maintains a precision improvement of at least three orders of magnitude. As shown in Table 2, this is largely due to the SReNet computed eigenpairs having smaller residuals (defined as $\|\mathcal{L}v - \lambda v\|_2$), indicating that SReNet can effectively solve for accurate eigenvalues and eigenfunctions simultaneously.

Additionally, Table 1 reveals that in the process of solving multiple eigenvalues, the errors for subsequent eigenvalues tend to be significantly higher than those for earlier ones. NeuralEF and NeuralSVD exhibit relatively stable error change, and But SReNet shows fluctuations (for instance, errors for $\lambda_2$ and $\lambda_3$ at dimension five are smaller than those for $\lambda_1$). This variability primarily arises because NeuralEF and NeuralSVD employ a uniform grid to acquire data points, whereas SReNet uses uniform random sampling. In high-dimensional problems, a uniform grid requires the

number of sampling points to satisfy an exponential form $num^D$, where $num$ is a grid number per dimension and $D$ is the operator dimension. However, uniform random sampling does not have this restriction.

## 4.2 SCHRÖDINGER OSCILLATOR EQUATION

The Schrödinger oscillator equation is a common problem in quantum mechanics, and its time-independent form is expressed as follows (Ryu et al., 2024; Griffiths & Schroeter, 2018):

$$-\frac{1}{2}\Delta\psi + V\psi = E\psi, \quad \text{in } \Omega = [0,1]^D, \tag{28}$$

where $\psi$ is the wave function, $\Delta$ represents the Laplacian operator indicating the kinetic energy term, $V$ is the potential energy within $\Omega$, and $E$ denotes the energy eigenvalue. This equation is formulated in natural units, simplifying the constants involved. Typically, the potential $V(x_1, \ldots, x_d) = \frac{1}{2}\sum_{k=1}^{d} x_k^2$ characterizes a multidimensional quadratic potential. The principal eigenvalue and corresponding eigenfunction are given by:

$$E_0 = \frac{d}{2}, \quad \psi_0(x_1, \ldots, x_d) = \prod_{k=1}^{d} \left(\frac{1}{\pi}\right)^{\frac{1}{4}} e^{-\frac{x_k^2}{2}}, \tag{29}$$

To validate our algorithm's capability in computing the principal eigenvalues in both low and high dimensions, this experiment focuses on calculating the ground states of the Schrödinger equation in one, two, and five dimensions, i.e. the smallest principal eigenvalues.

Table 3: Absolute error comparison for the principal eigenvalues of oscillator operators across algorithms. The first column lists the operator dimensions. The most accurate method is in bold.

| Method | PMNN | NeuralEF | NeuralSVD | SReNet |
|--------|------|----------|-----------|--------|
| Dim = 1 | 1.17e-6 | 2.57e-2 | 2.53e-2 | **3.62e-7** |
| Dim = 2 | 9.07e-5 | 7.55e-2 | 4.01e-1 | **2.35e-6** |
| Dim = 5 | 3.92e-1 | 3.97e-1 | 4.37e+0 | **3.23e-1** |

Firstly, as shown in Table 3, the SReNet algorithm achieves significantly higher precision than existing algorithms in computing the principal eigenvalues of the oscillator operator. Furthermore, the accuracy of SReNet surpasses that of PMNN. Both are designed based on the concept of the power method. When solving for the principal eigenvalue, the deflation projection loss may be considered inactive. This outcome suggests that the filter function significantly enhances the accuracy.

## 4.3 FOKKER-PLANCK EQUATION

The Fokker-Planck equation is central to statistical mechanics and is extensively applied across diverse fields such as thermodynamics, particle physics, and financial mathematics (Yang et al., 2023; Jordan et al., 1998; Frank, 2005). It can be mathematically formulated as follows:

$$-\Delta v - V \cdot \nabla v - \Delta V v = \lambda v, \quad \text{in } \Omega = [0, 2\pi]^D, \tag{30}$$

where the potential $V(x) = \sin\left(\sum_{i=1}^{d} c_i \cos(x_i)\right)$ is a potential function with each coefficient $c_i$ varying within $[0.1, 1]$, $\lambda$ the eigenvalue, and $v$ the eigenfunction. When the boundary conditions are periodic, the smallest eigenvalue is $\lambda = 0$, with the associated eigenfunction described by:

$$v(x) = e^{-V(x)}. \tag{31}$$

The eigenvalue at zero significantly impacts the numerical stability of the algorithm during iterative processes. To validate our algorithm's performance when the eigenvalue is zero, we compute the principal eigenvalues of the one and two dimensional Fokker-Planck equation as cases.

As indicated in Table 4, the SReNet algorithm significantly outperforms existing methods in computing the zero eigenvalues of the Fokker-Planck operator, effectively solving for cases where the

Table 4: Absolute error comparison for the principal eigenvalues of Fokker-Planck operators across algorithms. The first column lists the operator dimensions. The most accurate method in bold.

| Method | PMNN | NeuralEF | NeuralSVD | SReNet |
|--------|------|----------|-----------|--------|
| Dim = 1 | 8.60e-1 | 5.21e-1 | 2.73e-1 | **5.27e-2** |
| Dim = 2 | 8.30e-1 | 8.45e-1 | 2.75e-1 | **1.03e-1** |

eigenvalue is zero. It is mainly due to the filter function, which performs a spectral transformation on the operator, converting the zero eigenvalue into other eigenvalues that are easier to calculate without changing the eigenvector. However, compared to the experimental results for the Harmonic and quantum harmonic oscillator equations, the performance with the Fokker-Planck Equation is less favorable. This is primarily due to the oscillatory nature of the eigenfunctions, which presents greater challenges for neural network approximation.

## 4.4 ABLATION EXPERIMENTS

Table 5: Comparison of different settings of SReNet for the 2-dimensional Harmonic eigenvalue problem. "*" means SReNet without deflation projection and filter function.

| | Index | $\lambda$ Absolute Error | Residual |
|--|-------|--------------------------|----------|
| SReNet | $(v_1, \lambda_1)$ | 1.42e-5 | 4.12e-3 |
| | $(v_2, \lambda_2)$ | 2.96e-1 | 1.24e+1 |
| | $(v_3, \lambda_3)$ | 4.17e-1 | 1.43e+1 |
| SReNet* | $(v_1, \lambda_1)$ | 1.42e-5 | 4.12e-3 |
| | $(v_2, \lambda_2)$ | 2.96e+1 | 7.09e-3 |
| | $(v_3, \lambda_3)$ | 2.97e+1 | 1.09e-2 |

We conducted ablation experiments further to validate the performance of deflation projection and filter function. The results of the ablation experiments are shown in Table 5. The results indicate that the absence of deflation projection and projection has a significant impact on the prediction of eigenvalues. Without them, SReNet* is unable to eliminate the influence of previously solved eigenfunctions, resulting in the calculation being limited to the first eigenpair only. The residuals indicate that the second and third eigenpairs computed by SReNet* are actually identical to the first. In addition, experiments detailing the performance of SReNet as a function of model depth, model width, and the number of points can be found in Appendix E.

## 5 CONCLUSIONS AND FUTURE WORK

In this paper, we introduced SReNet, a learning-based method designed for solving operator eigenvalue problems. Our experiments demonstrate that SReNet achieves the highest accuracy compared to existing algorithms across a variety of operator eigenvalue problems. For future works, there are several key areas worth mentioning: 1. Algorithm optimization for specific operator structures, such as selecting more suitable deflation projections and filter functions based on the structure of the operator. 2. Designing better point distribution strategies, for example, adapting point placement based on boundary conditions and residual distribution. 3. Integrating other computational mathematics techniques to optimize the iterative process, such as incorporating matrix preconditioning technologies. We believe that neural network-based algorithms for solving operator eigenvalue problems hold tremendous potential for real-world applications and represent a crucial direction for future development.

## 6 CODE OF ETHICS AND ETHICS STATEMENT

This paper adheres to the ICLR Code of Ethics. The study aims to develop a more accurate learning-based method for solving operator eigenvalue problems. It does not involve human subjects, personal data, or sensitive information that could raise concerns regarding privacy, security, or fairness. Additionally, no potential conflicts of interest, legal compliance issues, or harmful applications were identified in this research.

## 7 REPRODUCIBILITY

To ensure reproducibility, we have included essential codes in the supplementary materials, covering dataset generation, the algorithm's source code, and performance evaluation scripts. However, it should be noted that the current version of the code lacks proper structure. If this paper is accepted, we are committed to reorganizing the code for better clarity. Additionally, Appendix C contains pseudocode for our algorithm. Furthermore, Appendix D contains a detailed description of our experimental setups.

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

## A    RELATED WORK

Recent advancements in applying neural networks to eigenvalue problems have shown promising results. Innovations such as Spectral Inference Networks (SpIN) (Pfau et al., 2018), which models eigenvalue problems as kernel problem optimizations solved via neural networks. Neural Eigenfunctions (NeuralEF) (Deng et al., 2022), which significantly reduces computational costs by optimizing the costly orthogonalization steps, are noteworthy. Neural Singular Value Decomposition (NeuralSVD) employs truncated singular value decomposition for low-rank approximation to enhance the orthogonality required in learning functions (Ryu et al., 2024).

Another class of algorithms originates from optimizing the Rayleigh quotient. The Deep Ritz Method (DRM) utilizes the Rayleigh quotient for computing the smallest eigenvalues, demonstrating significant potential (Yu et al., 2018). Several studies have employed the Rayleigh quotient to construct variation-free functions, achieved through Physics-Informed Neural Networks (PINNs) (Ben-Shaul et al., 2023; 2020). Extensions of this approach include enhanced loss functions with regularization terms to improve the learning accuracy of the smallest eigenvalues (Jin et al., 2022). Additionally, Han et al. (2020) reformulate the eigenvalue problem as a fixed-point problem of the semigroup flow induced by the operator, solving it using the diffusion Monte Carlo method. The Power Method Neural Network (PMNN) integrates the power method with PINNs, using an iterative process to approximate the exact eigenvalues (Yang et al., 2023) closely. While PMNN has proven effective in solving for a single eigenvalue (Yang et al., 2023), it has yet to be developed for computing multiple distinct eigenvalues simultaneously.

Furthermore, in the field of computational chemistry, research on specialized model architectures for specific operators, such as the Hamiltonian, focuses on developing novel neural network ansatzes (Carleo & Troyer, 2017; Schütt et al., 2017; Choo et al., 2020; Pfau et al., 2020; Hermann et al., 2020; Gerard et al., 2022; Hermann et al., 2023). These architectures are designed to embed physical inductive biases better, enhancing expressivity. Additionally, there are studies employing neural networks for Quantum Monte Carlo (QMC) methods to tackle related problems in quantum chemistry (Cuzzocrea et al., 2020; Entwistle et al., 2023; Pfau et al., 2023). Operators in specific scientific domains often exhibit unique structures and are associated with prior knowledge. One of our future research directions is to optimize our algorithm based on this information to achieve better performance in these tasks.

## B    BACKGROUND KNOWLEDGE AND RELEVANT ANALYSIS

### B.1    CONVERGENCE ANALYSIS OF THE POWER METHOD

Suppose $\boldsymbol{A} \in \mathbb{R}^{n \times n}$ and $\boldsymbol{V}^{-1}\boldsymbol{A}\boldsymbol{V} = \text{diag}(\lambda_1, \ldots, \lambda_n)$ with $\boldsymbol{V} = [\boldsymbol{v}_1 \quad \cdots \quad \boldsymbol{v}_n]$. Assume that $|\lambda_1| > |\lambda_2| \geq \cdots \geq |\lambda_n|$. The pseudocode for the power method is shown below (Golub & Van Loan, 2013):

---

**Algorithm 1:** Power method for finding the largest principal eigenvalue of the matrix $A$

---

**1** **Given** $\boldsymbol{A} \in \mathbb{R}^{n \times n}$ an $n \times n$ matrix, an arbitrary unit vector $x^{(0)} \in \mathbb{R}^n$, the maximum number of iterations $k_{\max}$, and the stopping criterion $\epsilon$.

**2** **for** $k = 1, 2, \ldots, k_{max}$ **do**

**3**   $\quad$ Compute $\boldsymbol{y}^{(k)} = \boldsymbol{A}\boldsymbol{x}^{(k-1)}$.

**4**   $\quad$ Normalize $\boldsymbol{x}^{(k)} = \frac{\boldsymbol{y}^{(k)}}{\|\boldsymbol{y}^{(k)}\|}$.

**5**   $\quad$ Compute the difference $\delta = \|\boldsymbol{x}^{(k)} - \boldsymbol{x}^{(k-1)}\|$.

**6**   $\quad$ **if** $\delta < \epsilon$ **then**

**7**   $\quad\quad$ Record the largest principal eigenvalue using the Rayleigh quotient,

$$\lambda^{(k)} = \frac{\langle \boldsymbol{x}^{(k)}, \boldsymbol{A}\boldsymbol{x}^{(k)} \rangle}{\langle \boldsymbol{x}^{(k)}, \boldsymbol{x}^{(k)} \rangle}.$$

$\quad\quad$ The stopping criterion is met, the iteration can be stopped.

Let us examine the convergence properties of the power iteration. If

$$\boldsymbol{x}^{(0)} = a_1 \boldsymbol{v}_1 + a_2 \boldsymbol{v}_2 + \cdots + a_n \boldsymbol{v}_n$$

and $\boldsymbol{v}_1 \neq 0$, then

$$\boldsymbol{A}^k \boldsymbol{x}^{(0)} = a_1 \lambda_1^k \left( \boldsymbol{v}_1 + \sum_{j=2}^{n} \frac{a_j}{a_1} \left( \frac{\lambda_j}{\lambda_1} \right)^k \boldsymbol{v}_j \right).$$

Since $\boldsymbol{x}^{(k)} \in \text{span}\{\boldsymbol{A}^k \boldsymbol{x}^{(0)}\}$, we conclude that

$$\text{dist}\left( \text{span}\{\boldsymbol{x}^{(k)}\}, \text{span}\{\boldsymbol{v}_1\} \right) = O\left( \left( \frac{\lambda_2}{\lambda_1} \right)^k \right).$$

It is also easy to verify that

$$|\lambda_1 - \lambda^{(k)}| = O\left( \left( \frac{\lambda_2}{\lambda_1} \right)^k \right).$$

Since $\lambda_1$ is larger than all the other eigenvalues in modulus, it is referred to as the largest principal eigenvalue. Thus, the power method converges if $\lambda_1$ is the largest principal and if $\boldsymbol{x}^{(0)}$ has a component in the direction of the corresponding dominant eigenvector $\boldsymbol{x}_1$ (Parlett & Poole, 1973; Wilkinson, 1965).

In practice, the effectiveness of the power method largely depends on the ratio $|\lambda_2|/|\lambda_1|$, as this ratio determines the convergence rate. Therefore, applying specific spectral transformations to the matrix to increase this ratio can significantly accelerate the convergence of the power method.

### B.2 DEFLATION PROJECTION DETAILS

Consider the scenario where we have determined the largest modulus eigenvalue, $\lambda_1$, and its corresponding eigenvector, $\boldsymbol{v}_1$, utilizing an algorithm such as the power method. These algorithms consistently identify the eigenvalue of the largest modulus from the given matrix along with an associated eigenvector. We ensure that the vector $\boldsymbol{v}_1$ is normalized such that $\|\boldsymbol{v}_1\|_2 = 1$. The task then becomes computing the subsequent eigenvalue, $\lambda_2$, of the matrix $\boldsymbol{A}$. A traditional approach to address this is through what is commonly known as a deflation procedure. This technique involves a rank-one modification to the original matrix, aimed at shifting the eigenvalue $\lambda_1$ while preserving all other eigenvalues intact. The modification is designed in such way that $\lambda_2$ emerges as the eigenvalue with the largest modulus in the adjusted matrix. Consequently, the power method can be reapplied to this updated matrix to extract the eigenvalue-eigenvector pair $\lambda_2, \boldsymbol{v}_2$.

When the invariant subspace requiring deflation is one-dimensional, consider the following Proposition 1. The propositions and proofs below are derived from Saad (2011) P90.

**Proposition 1.** *Let $\boldsymbol{v}_1$ be an eigenvector of $\boldsymbol{A}$ of norm 1, associated with the eigenvalue $\lambda_1$ and let $\boldsymbol{A}_1 \equiv \boldsymbol{A} - \sigma \boldsymbol{v}_1 \boldsymbol{v}_1^H$. Then the eigenvalues of $\boldsymbol{A}_1$ are $\tilde{\lambda}_1 = \lambda_1 - \sigma$ and $\tilde{\lambda}_j = \lambda_j, j = 2, 3, \ldots, n$. Moreover, the Schur vectors associated with $\tilde{\lambda}_j, j = 1, 2, 3, \ldots, n$ are identical with those of $\boldsymbol{A}$.*

*Proof.* Let $\boldsymbol{AV} = \boldsymbol{VR}$ be the Schur factorization of $\boldsymbol{A}$, where $\boldsymbol{R}$ is upper triangular and $\boldsymbol{V}$ is orthonormal. Then we have

$$\boldsymbol{A}_1 \boldsymbol{V} = \left[ \boldsymbol{A} - \sigma \boldsymbol{v}_1 \boldsymbol{v}_1^\top \right] \boldsymbol{V} = \boldsymbol{VR} - \sigma \boldsymbol{v}_1 \boldsymbol{e}_1^\top = \boldsymbol{V}[\boldsymbol{R} - \sigma \boldsymbol{e}_1 \boldsymbol{e}_1^\top].$$

Here, $\boldsymbol{e}_1$ is the first standard basis vector. The result follows immediately. $\square$

According to Proposition 1, once the eigenvalue $\lambda_1$ and eigenvector $\boldsymbol{v}_1$ are known, we can define the deflation projection matrix $\boldsymbol{P}_1 = \boldsymbol{I} - \lambda_1 \boldsymbol{v}_1 \boldsymbol{v}_1^\top$ to compute the remaining eigenvalues and eigenvectors.

When deflating with multiple vectors, let $\boldsymbol{q}_1, \boldsymbol{q}_2, \ldots, \boldsymbol{q}_j$ be a set of Schur vectors associated with the eigenvalues $\lambda_1, \lambda_2, \ldots, \lambda_j$. We denote by $\boldsymbol{Q}_j$ the matrix of column vectors $\boldsymbol{q}_1, \boldsymbol{q}_2, \ldots, \boldsymbol{q}_j$. Thus, $\boldsymbol{Q}_j \equiv [\boldsymbol{q}_1, \boldsymbol{q}_2, \ldots, \boldsymbol{q}_j]$ is an orthonormal matrix whose columns form a basis of the eigenspace associated with the eigenvalues $\lambda_1, \lambda_2, \ldots, \lambda_j$. An immediate generalization of Proposition 1 is the following (Saad, 2011) P94.

**Proposition 2.** *Let $\boldsymbol{\Sigma}_j$ be the $j \times j$ diagonal matrix $\boldsymbol{\Sigma}_j = diag(\sigma_1, \sigma_2, \ldots, \sigma_j)$, and $\boldsymbol{Q}_j$ an $n \times j$ orthogonal matrix consisting of the Schur vectors of $\boldsymbol{A}$ associated with $\lambda_1, \ldots, \lambda_j$. Then the eigenvalues of the matrix*

$$\boldsymbol{A}_j \equiv \boldsymbol{A} - \boldsymbol{Q}_j \boldsymbol{\Sigma}_j \boldsymbol{Q}_j^\top,$$

*are $\tilde{\lambda}_i = \lambda_i - \sigma_i$ for $i \leq j$ and $\tilde{\lambda}_i = \lambda_i$ for $i > j$. Moreover, its associated Schur vectors are identical with those of $\boldsymbol{A}$.*

*Proof.* Let $\boldsymbol{AU} = \boldsymbol{UR}$ be the Schur factorization of $\boldsymbol{A}$. We have

$$\boldsymbol{A}_j \boldsymbol{U} = \left[ \boldsymbol{A} - \boldsymbol{Q}_j \boldsymbol{\Sigma}_j \boldsymbol{Q}_j^\top \right] \boldsymbol{U} = \boldsymbol{UR} - \boldsymbol{Q}_j \boldsymbol{\Sigma}_j \boldsymbol{E}_j^\top,$$

where $\boldsymbol{E}_j = [\boldsymbol{e}_1, \boldsymbol{e}_2, \ldots, \boldsymbol{e}_j]$. Hence

$$\boldsymbol{A}_j \boldsymbol{U} = \boldsymbol{U} \left[ \boldsymbol{R} - \boldsymbol{E}_j \boldsymbol{\Sigma}_j \boldsymbol{E}_j^\top \right]$$

and the result follows. $\square$

According to Proposition 2, if $\boldsymbol{A}$ is a normal matrix and the eigenvalues $\lambda_1, \ldots, \lambda_j$ along with their corresponding eigenvectors $\boldsymbol{v}_1, \ldots, \boldsymbol{v}_j$ are known, we can construct the deflation projection matrix $\boldsymbol{P}_j = \boldsymbol{I} - \boldsymbol{V}_j \boldsymbol{\Sigma}_j \boldsymbol{V}_j^\top$ to compute the remaining eigenvalues and eigenvectors. Here, $\boldsymbol{\Sigma}_j = \text{diag}(\sigma_1, \sigma_2, \ldots, \sigma_j)$ and $\boldsymbol{V}_j = [\boldsymbol{v}_1, \boldsymbol{v}_2, \ldots, \boldsymbol{v}_j]$.

### B.3 FILTERING TECHNIQUE

The primary objective of filtering techniques is to manipulate the eigenvalue distribution of a matrix through spectral transformations (Saad, 2011). This enhances specific eigenvalues of interest, facilitating their recognition and computation by iterative solvers. Filter transformation functions, $F(x)$, typically fall into two categories:

1. Polynomial Filters, expressed as $P(x)$, such as the Chebyshev filter (Miao & Wu, 2021; Banerjee et al., 2016).

2. Rational Function Filters, often denoted as $P(x)/Q(x)$, such as the shift-invert method (Van Beeumen, 2015; Watkins, 2007). Below we describe this strategy in detail.

**Shift-Invert Strategy** The shift-invert strategy applies the transformation $(A - \sigma I)^{-1}$ to the matrix $A$, where $\sigma$ is a scalar approximating a target eigenvalue, termed as shift. This operation transforms each eigenvalue $\lambda$ of $A$ into $\frac{1}{\lambda - \sigma}$, amplifying those eigenvalues close to $\sigma$ in the transformed matrix, making them larger and more distinguishable (Watkins, 2007).

For instance, consider the power method, where the convergence rate is primarily governed by the ratio of the matrix's largest modulus eigenvalue to its second largest. Suppose matrix $A$ has three principal eigenvalues: $\lambda_1 = 10$, $\lambda_2 = 3$, and $\lambda_3 = 2$. Our objective is to compute $\lambda_1$, the largest eigenvalue. In the original matrix $A$, the convergence rate of the power method hinges on the spectral gap ratio, defined as:

$$\text{Spectral Gap Ratio} = \frac{\lambda_1}{\lambda_2} \approx 3.33$$

Applying the shift-invert transformation with $\sigma = 9.5$ strategically selected close to $\lambda_1$, the new eigenvalues $\mu$ are recalculated as:

$$\mu_i = \frac{1}{\lambda_i - \sigma}$$

This results in transformed eigenvalues:

$$\mu_1 = 2, \quad \mu_2 \approx -0.133, \quad \mu_3 \approx -0.125$$

Under this transformation, $\mu_1 = 2$ emerges as the dominant eigenvalue in the new matrix, with the other eigenvalues significantly smaller. Consequently, the new spectral gap ratio escalates to:

$$\text{New Spectral Gap Ratio} = \frac{2}{0.133} \approx 15.04$$

This enhanced spectral gap notably accelerates the convergence of the power method in the new matrix configuration.

Filtering techniques are often synergized with techniques like the implicit restarts of Krylov algorithms (Watkins, 2007; Golub & Van Loan, 2013), employing matrix operation optimizations to minimize the computational demands of evaluating matrix functions. These methods enable more precise localization and computation of multiple eigenvalues spread across the spectral range, particularly vital in physical (Salas et al., 2015; Banerjee et al., 2016) and materials science (Kohn, 1999) simulations where these eigenvalues frequently correlate with the system's fundamental properties (Winkelmann et al., 2019).

## C   ALGORITHM PSEUDOCODE

---

**Algorithm 2:** Spectral Refined Network

---

1 **Given**: $N$ (number of sampling points), $L$ (number of eigenvalues/eigenfunctions to compute), learning rate $\eta$, convergence threshold $\epsilon$, maximum iterations $K_{\max}$, PDE operator $\mathcal{L}$ over domain $\Omega \subset \mathbb{R}^D$.

2 Randomly sample $N$ points $\{\boldsymbol{x}_j\}_{j=1}^N$ in $\Omega$ to form dataset $S$.

3 Initialize computed eigenvalues list $\tilde{\Lambda} = [\,]$, eigenfunctions matrix $Q = [\,]$ and set iteration counter $k = 0$.

4 Set $Q_i = [\,], \Sigma_i = [\,], \tilde{\lambda}_i = 0, i = 0, \ldots, L - 1$.

5 Randomly initialize neural network parameters $\theta_i^0$.

6 **while** *not converged and $k < K_{max}$* **do**

7 $\quad$ For each $\boldsymbol{x}_j \in S$, compute neural network output:

$$\tilde{v}_i^k(\boldsymbol{x}_j) = NN_{\mathcal{L}}(\boldsymbol{x}_j; \theta_i^k), \quad i = 1, \ldots, L.$$

8 $\quad$ For each $\boldsymbol{x}_j \in S$, compute the update vector:

$$\tilde{u}_i^k(\boldsymbol{x}_j) = \frac{\mathcal{L}_i' \tilde{v}_i^k(\boldsymbol{x}_j)}{\left\| \mathcal{L}_i' \tilde{v}_i^k(\boldsymbol{x}_j) \right\|}, \quad \mathcal{L}_i' = \mathcal{L} - Q_{i-1}\Sigma_{i-1}Q_{i-1}^\top - \tilde{\lambda}_i I, \quad i = 1, \ldots, L.$$

9 $\quad$ Compute loss function:

$$\text{Loss}_{\mathcal{L}'}^{PM}(\theta_i^k) = \frac{1}{N} \sum_{j=1}^N \left[ \tilde{v}_i^{k-1}(\boldsymbol{x}_j) - \tilde{u}_i^k(\boldsymbol{x}_j) \right]^2, \quad i = 1, \ldots, L.$$

10 $\quad$ Update neural network parameters using gradient descent:

$$\theta_i^{k+1} = \theta_i^k - \eta \nabla_{\theta_i} \text{Loss}, \quad i = 1, \ldots, L.$$

11 $\quad$ **for** *i = 1 to L* **do**

12 $\quad\quad$ **if** $Loss_{\mathcal{L}'}^{PM}(\theta_i^k) < loss_{mini}$ **then**

13

$$loss_{mini} = \text{Loss}_{\mathcal{L}'}^{PM}(\theta_i^k).$$

14 $\quad\quad$ Update approximate eigenfunction and eigenvalue:

$$\tilde{v}_i = \tilde{v}_i^k, \quad \tilde{\lambda}_i = \frac{(\tilde{v}_i^k)^\top \mathcal{L}\tilde{v}_i^k}{(\tilde{v}_i^k)^\top \tilde{v}_i^k}, \quad i = 1, \ldots, L.$$

15 $\quad$ **if** $loss_{mini} < \epsilon, \quad i = 1, \ldots, L$ **then**

16 $\quad\quad$ Convergence achieved. Break the loop.

17 $\quad$ Else Update deflation projection and filter function

$$Q_i = [\tilde{v}_1, \tilde{v}_2, \ldots, \quad \tilde{v}_i], \quad \Sigma_i = \text{diag}(\tilde{\lambda}_1, \tilde{\lambda}_2, \ldots, \tilde{\lambda}_i), \quad i = 1, \ldots, L - 1.$$

18 $\quad$ Set $k = k + 1$ and continue.

19 **if** $k < K_{max}$ **then**

20 $\quad$ **Output**: eigenvalues diagonal matrix $\tilde{\Lambda} = [\tilde{\lambda}_1, \tilde{\lambda}_2, \ldots, \tilde{\lambda}_L]$ and eigenfunctions matrix $Q = [\tilde{v}_1, \tilde{v}_2, \ldots, \quad \tilde{v}_L]$.

21 **else**

22 $\quad$ **Output**: The maximum number of iterations has been reached, and the solution has not converged and failed .

---

## D  DETAILS OF EXPERIMENTAL SETUP

### D.1  EXPERIMENTAL ENVIRONMENT

To ensure consistency in our evaluations, all comparative experiments were conducted under uniform computing environments. Specifically, the environments used are detailed as follows:

- CPU: 72 vCPU AMD EPYC 9754 128-Core Processor
- GPU: NVIDIA GeForce RTX 4090D (24GB)

### D.2  EXPERIMENTAL PARAMETERS

- Neuralsvd and Neuralef:
    - Optimizer: rmsprop with a learning rate scheduler.
    - Learning rate: 1e-4, batch size: 128
    - Neural Network Architecture: layers = [128,128,128]
    - Laplacian regularization set to 0.01, with evaluation frequency every 10000 iterations.
    - Fourier feature mapping enabled with a size of 1024 and scale of 0.1.
    - Neural network structure: hidden layers of 128,128,128 using softplus as the activation function.
    - For the 1-dimensional problem, the number of points is $20,000$, with $400,000$ iterations. For the 2-dimensional problem, the number of points is $40,000 = 200 \times 200$, also with $400,000$ iterations. For the 5-dimensional problem, the number of points is $59,049 = 9^5$, with $500,000$ iterations.
- SReNet
    - Optimizer: Adam
    - Learning rate: 1e-4
    - Neural Network Architecture: Assuming d is the dimension of the problem. For d = 1 or 2, layers = [d, 20, 20, 20, 20, 1] (For Harmonic operator d=2, layers = [d, 20, 20, 20, 1]). For d=5, layers = [d, 40, 40, 40, 40, 1]. For else case, layers = [d, 40, 40, 40, 40, 1].
    - For the 1-dimensional problem, the number of points is $20,000$, with $400,000$ iterations. For the 2-dimensional problem, the number of points is $40,000 = 200 \times 200$, also with $400,000$ iterations. For the 5-dimensional problem, the number of points is $59,049 = 9^5$, with $500,000$ iterations.

### D.3  ERROR METRICS

- Absolute Error:
  We employ absolute error to estimate the bias of the output eigenvalues of the model:

$$\text{Absolute Error} = |\tilde{\lambda} - \lambda|. \tag{32}$$

  Here $\tilde{\lambda}$ represents the eigenvalue predicted by the model, while $\lambda$ denotes the true eigenvalue.

- Residual Error:
  To further analyze the error in eigenpair $(\tilde{v}, \tilde{\lambda})$ predictions, we use the following metric:

$$\text{Residual Error} = ||\mathcal{L}\tilde{v} - \tilde{\lambda}\tilde{v}||_2. \tag{33}$$

  Here, $\tilde{v}$ represents the eigenfunction predicted by the model. When $\tilde{\lambda}$ is the true eigenvalue and $\tilde{v}$ is the true eigenfunction, the Residual Error equals 0.

# E ANALYSIS OF HYPERPARAMETERS

**Model Depth**:

Table 6: Consider the 2-dimensional Harmonic problem, with the fixed layer width of 20, and compare the performance of SReNet at different model layers. Other experimental details are the same as Appendix D.2.

| Layer | Index | $\lambda$ Absolute Error | Residual |
|---|---|---|---|
| 3 | $(v_1, \lambda_1)$ | 1.02e-5 | 4.56e-3 |
|   | $(v_2, \lambda_2)$ | 3.04e-2 | 2.56e+1 |
|   | $(v_3, \lambda_3)$ | 6.76e-2 | 6.99e+1 |
|   | $(v_4, \lambda_4)$ | 1.00e-1 | 2.12e+3 |
| 4 | $(v_1, \lambda_1)$ | 1.42e-5 | 4.12e-3 |
|   | $(v_2, \lambda_2)$ | 2.96e-1 | 1.24e+1 |
|   | $(v_3, \lambda_3)$ | 4.17e-1 | 1.43e+1 |
|   | $(v_4, \lambda_4)$ | 2.00e+1 | 2.17e+5 |
| 5 | $(v_1, \lambda_1)$ | 4.36e-6 | 4.12e-3 |
|   | $(v_2, \lambda_2)$ | 8.63e-1 | 3.12e+1 |
|   | $(v_3, \lambda_3)$ | 1.98e+0 | 1.58e+3 |
|   | $(v_4, \lambda_4)$ | 8.94e+1 | 2.09e+3 |
| 6 | $(v_1, \lambda_1)$ | 1.06e-5 | 9.56e-3 |
|   | $(v_2, \lambda_2)$ | 8.21e-1 | 2.00e+1 |
|   | $(v_3, \lambda_3)$ | 1.17e+0 | 9.90e+3 |
|   | $(v_4, \lambda_4)$ | 3.81e+1 | 7.53e+4 |

**Model Width**:

Table 7: Consider the 2-dimensional Harmonic problem, with the fixed layer depth of 3, and compare the performance of SReNet at different model widths. Other experimental details are the same as Appendix D.2.

| Width | Index | $\lambda$ Absolute Error | Residual |
|---|---|---|---|
| 10 | $(v_1, \lambda_1)$ | 1.68e-6 | 1.26e-3 |
|    | $(v_2, \lambda_2)$ | 3.82e-1 | 2.36e+0 |
|    | $(v_3, \lambda_3)$ | 7.54e-1 | 1.20e+2 |
|    | $(v_4, \lambda_4)$ | 1.71e-1 | 2.49e+3 |
| 20 | $(v_1, \lambda_1)$ | 1.42e-5 | 4.12e-3 |
|    | $(v_2, \lambda_2)$ | 2.96e-1 | 1.24e+1 |
|    | $(v_3, \lambda_3)$ | 4.17e-1 | 1.43e+1 |
|    | $(v_4, \lambda_4)$ | 2.00e+1 | 2.17e+5 |
| 30 | $(v_1, \lambda_1)$ | 3.26e-5 | 2.25e-2 |
|    | $(v_2, \lambda_2)$ | 1.50e+0 | 2.10e+1 |
|    | $(v_3, \lambda_3)$ | 1.59e+0 | 8.21e+3 |
|    | $(v_4, \lambda_4)$ | 3.52e+2 | 2.77e+5 |
| 40 | $(v_1, \lambda_1)$ | 1.57e-5 | 2.06e-2 |
|    | $(v_2, \lambda_2)$ | 2.67e+0 | 5.03e+1 |
|    | $(v_3, \lambda_3)$ | 7.93e+1 | 5.76e+3 |
|    | $(v_4, \lambda_4)$ | 1.50e+2 | 1.47e+4 |

**The Number of Points**:

Table 8: Consider the 2-dimentional Harmonic problem and compare the performance of SReNet at different number of points. Other experimental details are same Appendix D.2.

| Number | Index | $\lambda$ Absolute Error | Residual |
|---|---|---|---|
| | $(v_1, \lambda_1)$ | 1.11e-5 | 3.19e-3 |
| 20000 | $(v_2, \lambda_2)$ | 1.25e+0 | 3.22e+0 |
| | $(v_3, \lambda_3)$ | 1.61e+0 | 1.27e+2 |
| | $(v_1, \lambda_1)$ | 4.40e-5 | 7.09e-3 |
| 30000 | $(v_2, \lambda_2)$ | 3.58e-1 | 2.71e+0 |
| | $(v_3, \lambda_3)$ | 1.70e-1 | 5.62e+1 |
| | $(v_1, \lambda_1)$ | 1.42e-5 | 4.12e-3 |
| 40000 | $(v_2, \lambda_2)$ | 2.96e-1 | 1.24e+1 |
| | $(v_3, \lambda_3)$ | 4.17e-1 | 1.43e+1 |
| | $(v_1, \lambda_1)$ | 4.94e-6 | 6.63e-3 |
| 50000 | $(v_2, \lambda_2)$ | 2.53e-1 | 2.46e+1 |
| | $(v_3, \lambda_3)$ | 3.73e-1 | 1.50e+3 |

The influence of model depth, model width, and the number of points on SReNet is illustrated in Tables 6, 7, and 8, respectively. Experimental results indicate that SReNet is relatively unaffected by changes in model depth and model width. However, it is significantly influenced by the number of points, with performance improving as more points are used.

