# OpenReview forum: "SReNet: Spectral Refined Network for Solving Operator Eigenvalue Problem"
_ICLR.cc/2025/Conference — ICLR 2025 Conference Withdrawn Submission_

### Official Review · Reviewer_1jXS · 2024-10-27

**Soundness:** 2
**Presentation:** 1
**Contribution:** 1
**Rating:** 3
**Confidence:** 3

**Summary:**

The authors propose a neural network based approach for solving operator eigenvalue problems. There is an interest in approximating  eigenvalues and eigenfunctions for a given operator. The authors  propose the Spectral Refined Network (SReNet) based on the power method.

**Strengths:**

- Highly relevant problem in a variety of disciplines
- Sound introduction of eigenvalue basis given the page restrictions

**Weaknesses:**

- Difficult to follow derivation of the suggested method and it is also unclear to me what the advantage is of a neural network based approach.
- Convergence behaviour not well understood
- Comparison to outside of learning based approaches missing
- Notation for the neural network as $NN_\mathcal{L}$ confusing if in the next line $N$ appears as the number of sampling points
- The authors introduce in equation (10) but the meaning of the term only gets clarified in equation (12) when the term based on the power method is introduced
- Are the results in Table 1 really absolute errors? What about relative errors? These could be much more meaningful as it is not clear whether the methods produce anything sensible.
- Most of the references are formatted poorly. What is the first reference supposed to be? DME Zurich is likely not an author. Capitalize the names of Krylov, Schur, Fokker, Planck, etc

**Questions:**

- What do the authors know about the performance of the method? How does it depend on the spectral gap between the two dominating eigenvalues?
- Filtering and deflation are essentially sequential in nature if you compare to a state-of-at-art Krylov based eigenvalues solver. Why is it even sensible to switch to a learning based procedure?

---

### Official Review · Reviewer_XSjt · 2024-11-03

**Soundness:** 1
**Presentation:** 1
**Contribution:** 1
**Rating:** 1
**Confidence:** 5

**Summary:**

This paper proposes a neural network based approach to solving operator eigenvalue problems. The main technical tools are derived from numerical linear algebra, particularly the power method and the deflation projection widely applied in solving matrix eigenproblems. The authors presented comparisons with existing approaches.

**Strengths:**

The operator eigenproblem has broad applications in the physics and engineering domains. Recent works have actively explored the possibility of using deep learning tools to solve such problems and overcome the limitations of numerical linear algebra tools. This paper attempted to incorporate numerical tools with deep learning, an important topic.

**Weaknesses:**

Though the paper presented evaluations to suggest the advantages of the proposed method, there are significant technical flaws from the framework and algorithm design to evaluation approaches. Therefore, the conclusions in the manuscript could be misleading.

1. **Formulation**: The paper extensively discussed and compared with previous works on neural eigensolvers, including NeuralEF [Deng2022], NeuralSVD [Ryu2024], and PMNN [Yang2023]. Though Figure 2 of the manuscript resembles Figure 1 of NeuralSVD,  this work does *NOT* provide a training objective (loss function) for neural network training, unlike NeuralEF or NeuralSVD.  Instead, the implementation requires an iterative optimization step [cf. Eq (12)] of two terms. However, the details, e.g., parameter design or the stability of the iterating procedure, were not discussed in the manuscript.
2. **Results & Evaluation**: From the manuscript (appendix included), it is unclear how the authors addressed the generalization issue. From a high level, the work is built upon discretizing the function by taking $N$ random samples, while the $N$ samples are fixed during training [cf. Algorithm 2, P19]. Notably, in Eq. (33), the residual error is computed from the same collection $N$ samples. In other words, the training procedure was designed to minimize the error on a collection of fixed samples, and the training error was reported as the residual, which causes overfitting and is problematic. The main challenge in eigenfunction computation is generalizing to the whole domain, which was not mentioned in the manuscript.
    - In D.2, the network architecture and optimization for SReNet differ from the other baselines (NeuralSVD and NeuralEF), making the comparison questionable.
    - The result in Figure 1 of the manuscript showing the comparison between the proposed approach and NeuralSVD is inconsistent with the result reported in NeuralSVD paper [Ryu2024, Figure 4].

The presentation of the manuscript is not very clear, particularly the implementation of Eq. (12), i.e.,  "iterative optimization." The notations in Section 3 are hard to follow. Typos including $\boldsymbol{x_j}$, which should be $\boldsymbol{x}_j$.

---
1. Zhijie Deng, Jiaxin Shi, and Jun Zhu. Neuralef: Deconstructing kernels by deep neural networks.
In International Conference on Machine Learning, pp. 4976–4992. PMLR, 2022.
2. J Jon Ryu, Xiangxiang Xu, HS Erol, Yuheng Bu, Lizhong Zheng, and Gregory W Wornell. Operator
SVD with neural networks via nested low-rank approximation. arXiv preprint arXiv:2402.03655, 2024.
3. Qihong Yang, Yangtao Deng, Yu Yang, Qiaolin He, and Shiquan Zhang. Neural networks based on
power method and inverse power method for solving linear eigenvalue problems. Computers &
Mathematics with Applications, 147:14–24, 2023.

**Questions:**

See the weakness part. In particular, it would be better if the authors could clarify where the performance gain comes from (suppose the evaluation issues have been fixed.)

Two specific questions are:
1) how does the approach address the generalization issue?
2) how much performance gain is due to the deflation approach? Or, more generally, the incorporation of numerical linear algebra and learning approaches.

---

### Official Review · Reviewer_V1oZ · 2024-11-04

**Soundness:** 2
**Presentation:** 2
**Contribution:** 1
**Rating:** 3
**Confidence:** 4

**Summary:**

This paper proposes a method to solve eigenvalue problems for differential operators by using neural networks to represent the eigenvectors.

**Strengths:**

The writing is clear and the methodology is explained clearly.

**Weaknesses:**

- The authors claim that eigenvalue problems suffer from the curse of dimensionality, however for the computation of first few eigenvalues is actually much more benign e.g. using sparse grids. So if the goal is to compute first few eigenfunctions, the authors should compare this method against existing sparse grid methods.

- The approach of the author is not completely distinct from the NeuralEF method: their minimization problem can be written as finding the critical points of $\sum_j v_i(x_j) [\mathcal{L} v_i] (x_j)$ if one expands (12).  In this light, the method can be viewed as a variant that seeks to accelerate the convergence of NeuralEF. The authors should comment on this connection.

**Questions:**

- Did the authors really mean to cite the reference (LeVeque, 2002), which is a book on finite volume methods, not finite element methods?

- Does the normalization by $|| \mathcal{L} v_i (x_j) ||$ refer to a division by a pointwise scalar? This is then different from the usual power method. Do the authors mean normalization by  $(\sum_j |\mathcal{L} v_i (x_j)|^2)^{1/2}$, or similar?

---

### Official Review · Reviewer_ZEZA · 2024-11-07

**Soundness:** 2
**Presentation:** 2
**Contribution:** 2
**Rating:** 3
**Confidence:** 3

**Summary:**

The paper proposes a learning-based approach for solving an operator eigenvalue problem. The difference from existing studies is to incorporate deflation and shift-and-invert preconditioning technique into the learning approach such that the neural solver can find top-L eigen pairs and accelerate the convergence. Experiments are conducted on several operator eigenvalue problems.

**Strengths:**

The strength of the paper is to improve the existing learning based approach to top eigen-pair computation for operators via the deflation and preconditioning. Extensive experiments are conducted.

**Weaknesses:**

There are several concerns.
1) It's unclear why Eq. (10) can find the largest eigenpair. The optimal value of the objective function in Eq. (10) is 0, which can be attained at any eigenpair.
2) There is no analysis on the sample complexity such that the optimal solution is guaranteed.
3) It's unclear how to understand Eq. (18) and Eq. (22) since they are a mix of operator and matrices.
4) Why can the inverse operation be omitted in Eq. (24) given the goal of amplifying the target eigenvalues?
5) What's the difference of the proposed algorithm from the work by Yang et al. 2023?
6) In Table 1, 9 orders of magnitude improvement occurred at the largest eigenpair, which doesn't require deflation at all. It's unclear how the proposed approach makes it.

**Questions:**

see above

---

### Note · Authors · 2024-11-17

I have read and agree with the venue's withdrawal policy on behalf of myself and my co-authors.